# A One-Week Elderberry Juice Intervention Augments the Fecal Microbiota and Suggests Improvement in Glucose Tolerance and Fat Oxidation in a Randomized Controlled Trial

**DOI:** 10.3390/nu16203555

**Published:** 2024-10-20

**Authors:** Christy Teets, Nancy Ghanem, Guoying Ma, Jagrani Minj, Penelope Perkins-Veazie, Sarah A. Johnson, Andrea J. Etter, Franck G. Carbonero, Patrick M. Solverson

**Affiliations:** 1Department of Nutrition and Exercise Physiology, Elson S Floyd College of Medicine, Washington State University, Spokane, WA 99202, USA; christy.teets@wsu.edu (C.T.); jagrani.minj@wsu.edu (J.M.); franck.carbonero@wsu.edu (F.G.C.); 2Department of Food Science and Human Nutrition, Colorado State University, Fort Collins, CO 80523, USA; nancy.ghanem@colostate.edu (N.G.); sarah.johnson@colostate.edu (S.A.J.); 3Plants for Human Health Institute, Department of Horticultural Science, North Carolina State University, Kannapolis, NC 28081, USA; gma@ncsu.edu (G.M.); penelope_perkins@ncsu.edu (P.P.-V.); 4Department of Nutrition and Food Science, University of Vermont, Burlington, VT 05405, USA; andrea.etter@uvm.edu

**Keywords:** berry anthocyanins, functional foods, obesity, fecal microbiome, substrate oxidation, meal tolerance test, insulin and glucose homeostasis

## Abstract

Obesity is a costly and ongoing health complication in the United States and globally. Bioactive-rich foods, especially those providing polyphenols, represent an emerging and attractive strategy to address this issue. Berry-derived anthocyanins and their metabolites are of particular interest for their bioactive effects, including weight maintenance and protection from metabolic aberrations. Earlier findings from small clinical trials suggest modulation of substrate oxidation and glucose tolerance with mediation of prospective benefits attributable to the gut microbiota, but mixed results suggest appropriate anthocyanin dosing poses a challenge. The objective of this randomized, placebo-controlled study was to determine if anthocyanin-dense elderberry juice (EBJ) reproduces glucoregulatory and substrate oxidation effects observed with other berries and if this is mediated by the gut microbiota. Overweight or obese adults (BMI > 25 kg/m^2^) without chronic illnesses were randomized to a 5-week crossover study protocol with two 1-week periods of twice-daily EBJ or placebo (PL) separated by a washout period. Each treatment period included 4 days of controlled feeding with a 40% fat diet to allow for comparison of measurements in fecal microbiota, meal tolerance testing (MTT), and indirect calorimetry between test beverages. Eighteen study volunteers completed the study. At the phylum level, EBJ significantly increased Firmicutes and Actinobacteria, and decreased Bacteroidetes. At the genus level, EBJ increased *Faecalibacterium*, *Ruminococcaceae*, and *Bifidobacterium* and decreased *Bacteroides* and lactic acid-producing bacteria, indicating a positive response to EBJ. Supporting the changes to the microbiota, the EBJ treatment significantly reduced blood glucose following the MTT. Fat oxidation also increased significantly both during the MTT and 30 min of moderate physical activity with the EBJ treatment. Our findings confirm the bioactivity of EBJ-sourced anthocyanins on outcomes related to gut health and obesity. Follow-up investigation is needed to confirm our findings and to test for longer durations.

## 1. Introduction

Over 70% of adults in the United States have overweight or obesity [1]. The most recent estimates indicate 42% of adults suffer from obesity, and this will increase to 48–55% by 2050 [2,3]. There are myriad and multi-faceted causes for obesity. Prospective dietary management of the cardiometabolic complications associated with obesity include dietary patterns which incorporate food sources rich in bioactive food constituents, such as the Mediterranean-style diet [4]. These dietary patterns include 5–10 daily servings of fruits and vegetables, which are dense sources of polyphenols that promote human health and longevity.

Anthocyanins are a diverse sub-class of flavonoids extensively researched for their health-promoting properties which includes the metabolic alterations associated with obesity including diabetes, dyslipidemia, and cardiovascular disease [5]. Additionally, findings spanning translational rodent studies to large, prospective cohort studies demonstrate protective effects of anthocyanin-rich berries against obesity-associated morbidities and mortality [6,7,8,9]. Some mechanisms of action of anthocyanin’s benefits include preventing the intestinal absorption of monosaccharides, enhanced cellular metabolism of adipose and muscle tissues, and modulation of the gut microbiome [10,11].

We previously demonstrated that 600 g of blackberries per day for 1 week both increases insulin sensitivity, as evidenced by a meal tolerance test, and also increases fat oxidation, demonstrated via a reduction in the respiratory quotient from 24 h of indirect calorimetry [12]. The same observations were not unequivocally reproduced by a mixed-berry intervention [13]. Therefore, anthocyanin type and daily dose require further scrutiny to determine which anthocyanins and at what threshold doses may offer benefit to weight management and glucose homeostasis. We recently studied elderberry juice due to its remarkably high density of cyanidin-based anthocyanins [14]. In that pilot study, we demonstrated tolerability of 177.5 g of elderberry juice daily for 1 week, and possible modulation of substrate oxidation. Additionally, we also confirmed the prospective anti-obesity effects of elderberry juice powder using a diet-induced obese rodent model [15]. In that 12-week study, rodents fed a 45% fat diet supplemented with 10% elderberry juice powder did not present an obese phenotype compared to the control group [15]. Main findings included a normalized growth curve and body composition and improved fasting blood insulin despite eating significantly more kilocalories and maintaining the same physical activity as control mice. In the gut microbiome, the elderberry intervention increased *Bifidobacterium*, *Akkermansia*, and *Anaeroplasma* relative abundances, suggesting the physical and metabolic improvements are at least partially driven by changes to the microbiome [15].

The objective of this current human study is to determine if the metabolic benefits observed with other anthocyanin-rich berries extend to elderberries. To accomplish this, we performed a short-term controlled feeding study in overweight but healthy adult volunteers in a randomized, placebo-controlled crossover clinical trial. Major study endpoints included blood glucose and insulin response to a high-carbohydrate meal challenge, indirect calorimetry, and profiling of the fecal microbiome.

## 2. Materials and Methods

### 2.1. Human Participants

Potential female and male study participants aged 22–75 years were recruited from the Spokane (WA, USA) metropolitan area using a rolling recruitment and follow-up approach, as we have described previously [14]. Study advertisements were circulated through several advertising venues including a Washington State University (WSU) ‘participate in research’ website, a neighborhood-based social media platform, posted flyers, and electronic forums. Interested participants were screened for eligibility with a health questionnaire, study application, blood chemistries, and anthropometrics. Volunteers were disqualified if they were pregnant or intended to become pregnant, had a body mass index below 25 kg/m^2^, had an allergy/intolerance to elderberries, a history of bariatric surgery or malabsorption diseases, restrictive dietary patterns, habitual tobacco use in the last 6 months, significant (10%) weight loss or gain in the preceding 2 months, cancer in the preceding 3 years, Crohn’s disease, diverticulitis, or other gastrointestinal issues, use of blood-thinning or other medications that could complicate participant safety or interfere with study outcomes, type 2 diabetes requiring management with prescription medication, fasting blood glucose above 125 mg/dL, or active alcoholism. Qualified participants were screened for atypical diet patterns using the ASA24 from the National Cancer Institute (Frederick, MD, USA). All enrolled participants provided written informed consent prior to study participation. They were instructed to follow their habitual diet pattern and cease taking dietary supplements beyond multivitamins, calcium, and vitamin D for the entire 5 weeks of the study, and not donate blood. They were provided with a list of high-polyphenol foods to avoid, including all types of edible berries, red grapes, cherries, plums, red apples, red cabbage, red radishes, red onions, eggplant, and darkly colored beans. They were asked to avoid these foods and products for the entire 5-week study. The study protocol was approved by the WSU Institutional Review Board (Pullman, WA, USA) under number 18597. The study protocol was registered on ClinicalTrials.gov (NCT 06626373). The study was conducted at the Department of Nutrition and Exercise Physiology (NEP) of WSU’s Elson S. Floyd College of Medicine in Spokane, WA, USA.

### 2.2. Study Design and Treatments

The present study design is similar to our previous pilot study with modifications [14]. Briefly, the study was randomized and placebo-controlled, with a crossover design that included two separate treatments. Each treatment period was 1 week in length and separated by a 3-week washout period. Participants were asked to consume 355 g of either 100% elderberry juice (EBJ) (River Hills Harvest, Hartsburg, MO, USA), or a flavor-and-color-matched placebo beverage (PL) every day for 1 week. The 355 g daily serving was estimated to provide 720 mg of cyanidin-3-glucoside equivalents (C3GE) per day, with the C3GE concentration in the elderberry juice measured by the total monomeric anthocyanin spectrophotometric assay [16], and later confirmed by LC using a method described previously (Table 1) [17]. The placebo beverage was prepared by North Carolina State University’s Food Innovation Lab (Kannapolis, NC, USA).

Participants were blinded to the treatment and sequence. They were randomized to sequence 1 (elderberry juice then placebo) or sequence 2 (placebo then elderberry juice) using the method of covariate adaptive randomization. With the rolling-recruitment nature of the study, after the first participant’s sequence was randomly assigned, each additionally enrolled participant was assigned the opposite sequence to ensure sequence assignment was balanced over the timespan of the recruitment period. Daily doses were measured into two 14-ounce beverage containers containing 177.5 g of elderberry juice or placebo, and participants were instructed to consume one container in the morning and one in the evening with meals. To minimize variability in physiological testing attributable to background diet, participants were provided a 100% investigator-controlled background diet for the last 4 days of each diet period leading up to physiological testing and biospecimen collections on day 8. The controlled diet was designed to have a macronutrient profile of 40% energy from carbohydrates, 40% from fat, and 20% from protein. The diets were designed using Food Processor, version 11.11, (ESHA, Beaverton, OR, USA). The menus were designed to match either a 2200-kcal diet or a 2800-kcal diet. Meals were provided as a 2-day menu and were consumed twice for a total of 4 consecutive days, with each menu on an alternate day. Menus were comprised of single-serving prepackaged, yet non-perishable (i.e., shelf-stable) food items. If a participant did not reach satiety with the provided foods alone, an extra snack containing ~200 kcal with the same macronutrient profile was provided. Participants were instructed to eat only the food provided to them, except for water. Of note, prepackaged foods are known to contain higher sodium content than fresh foods, and we maintained sodium amounts below 5000 mg per day. The daily caloric needs for each participant were estimated according to the Mifflin-St. Jeor equation to determine daily caloric requirements, and each participant was prescribed to the appropriate meal and snack plan that would maintain body weight. Treatments and meals were consumed offsite as part of the participants’ usual work/life schedule. Participants were provided with a 7-day supply of juice or placebo and instructed to store their treatment containers at 4 °C until consumption. They completed a daily questionnaire to report test beverage, menu, and snack consumption, and any medication use or acute illness. Compliance was assessed from returned food and beverage containers and questionnaires collected on the morning of the 8th day of the respective treatment period, coinciding with their test day.

### 2.3. Participant Testing with Meal and Exercise Challenges

Our meal tolerance test (MTT) method has been described before [14]. We employed a similar method here with minor changes. On the morning of the eighth day of each 7-day treatment period, participants reported to NEP’s clinical testing laboratory between 7 and 8 a.m. after a minimum 12-h fast for combined indirect calorimetry (IC) and MTT testing. Following a weigh-in, collection of empty food and beverage containers, completed daily questionnaires, and a general health check-in, an intravenous catheter was placed in the antecubital vein by a study nurse. After two baseline blood samplings separated by at least 10 min, participants were given 10 min to consume a challenge meal consisting of 80 g toaster waffles, 80 g pancake syrup, and 355 g EBJ or placebo beverage, according to the preceding week’s respective treatment assignment. A high-carbohydrate challenge meal was used in lieu of a sucrose solution to model a more representative blood glucose and insulin incursions following food consumption. The test meal serving sizes were designed to provide at least 60 g of sugar from whole food sources (waffles and syrup), in addition to the sugar content of the respective test beverages (Table 2). The placebo beverage was formulated to contain 11% sugar, a standard concentration for elderberry juice described in the code of federal regulations [18]. The sugar content of both test beverages was measured using a standard spectrophotometric assay for sucrose, D-fructose, and D-glucose following the manufacturer’s protocol (Megazyme, Bray, Co., Wicklow, Ireland).

Following consumption of the challenge meal, participants were made comfortable on a padded examination table in a recumbent position under a TrueOne 2300 metabolic cart canopy system (ParvoMedics, Provo, UT, USA) to measure respiratory gases. After a steady state was reached, resting measurements were collected for 3 h from the time of the first bite of food. Following the final blood sampling, discharge of the intravenous catheter, and a 15 to 30-min break, participants walked on a treadmill for 30 min at 3 miles per hour while IC was assessed using the same metabolic cart system.

### 2.4. Blood Glucose and Plasma Insulin

To generate 3 h blood glucose and insulin response curves, blood sampling occurred every thirty minutes from the first bite of the challenge meal for 3 h, totaling 8 blood sampling timepoints: −15, 0, 30, 60, 90, 120, 150, and 180 min. Blood was collected into two 8 mL serum tubes, clotted, centrifuged, aliquoted, and stored at −80 °C until analysis. Insulin concentration was measured using the Millipore ELISA Kit for Human Insulin (Millipore Sigma, Burlington, MA, USA; EZHI-14K). Each sample was assessed in duplicate, and absorbance was measured at 450 nm using a BioTek SYNERGY H1 microplate reader (Agilent, Santa Clara, CA, USA). Plasma glucose concentrations were analyzed using a Glucose Colorimetric Assay Kit (Cayman Chemicals, Ann Arbor, MI, USA; #10009582) in duplicate with absorbance measured at 500 nm.

### 2.5. Gut Microbiota

To determine the effect of EBJ consumption on the gut microbiome composition, baseline (between days 3 and 4, immediately preceding controlled feeding) stool samples (EBJ1 or PL1) and post-intervention (between days 7 and 8, after 4 days of controlled feeding) stool samples (EBJ2 or PL2) were collected during each diet period. Directly after the stool samples were delivered, a fresh fecal slurry was prepared inside a sterile filter bag (Filtra-Bag, Labplas, Sainte-Julie, QC, Canada) containing phosphate buffered saline, pH 7.0, added with 10% glycerol as a cryoprotectant. The mixture was stomacher-mixed for 4 min at 300 rpm (Stomacher™ 400 Circulator, Seward™) and transferred inside an anaerobic chamber (90% N_2_, 5% CO_2_ and 5% H_2_, Anaerobic Gas Infuser, COY Laboratory Products, Inc., Grass Lake, MI, USA) and aliquoted into 10 mL volumes before being snap frozen in liquid nitrogen. The fecal slurries were stored at −80 °C until use. Fecal DNA was extracted using the QIAamp Fast DNA Stool Mini Kit (Qiagen, Hilden, Germany) according to the manufacturer’s instructions with addition of a recommended bead-beating step [19]. Approximately 200 mg of fecal samples were bead-beated in extraction buffer with a TissueLyser LT (Qiagen, Germany), heated at 95 °C for 10 min, centrifuged, and treated with Proteinase K. DNA was eluted in 100 μL Buffer ATE. Extracted DNA was quantified using Qubit (Invitrogen, Carlsbad, CA, USA). The 16S rRNA gene amplicon library preparation and sequencing were performed by Microbiome Insights (Vancouver, BC, Canada). For library preparation, bacterial 16S rRNA genes were PCR-amplified with dual-barcoded primers targeting the V4 region (515F 5′-GTGCCAGCMGCCGCGGTAA-3′, and 806R 5′-GGACTACHVGGGTWTCTAAT-3′), as per the protocol of Kozich et al. [20]. Amplicons were sequenced with an Illumina MiSeq using the 300-bp paired-end kit (v.3). Sequences were demultiplexed and barcodes were removed prior to sequence analysis with the Mothur software, v1.47.0 [21]. Sequence quality control, trimming, chimera removal and denoising were performed with vsearch and uchime [22,23]. Taxonomy was assigned using the SILVA database [24]. Reads were rarefied to a sampling depth of the lowest number observed for one sample prior to statistical and diversity analysis. Diversity metrics of the fecal samples were calculated using PAST software, v4.17 [25].

### 2.6. Calculations and Statistics

The calculations and statistics are like those described in our pilot study [14]. The respiratory gas measurements were used to calculate grams of carbohydrate (CHO) and fat oxidized for energy, the respiratory quotient (CO_2_ produced/O_2_ consumed), and energy expenditure in kilocalories. CHO and fat oxidation, energy expenditure, and average RQ were calculated on a minute-by-minute basis. Data from the initial 0–10 min that did not fall within steady state were removed. Steady state was defined as achieving a ≤10% coefficient of variation (CV) for a 5-min period using VO_2_ and VCO_2_ [26]. Respiratory gases were also measured during the 30 min treadmill walk to calculate the same outcomes.

Serum glucose and insulin fasting values (−15 and 0 min) were averaged and seven-point response curves for each variable were created from the 3 h MTT for both diet periods. Incremental area under the concentration curve (iAUC) was calculated for each curve using the central Riemann-sum theorem. Serum glucose concentration is reported as mg glucose per dL serum, and insulin concentration is reported as µU insulin per mL serum. Serum glucose iAUC is reported as mg·minute per dL, and serum insulin iAUC is reported as µU·minute per mL.

Linear mixed modeling was employed to test for statistically significant differences between the EBJ and PL beverage treatments using PROC MIXED repeated measures analysis of covariance in SAS version 9.4 (Cary, NC, USA). Response variables for each treatment period were repeated on volunteer; and the best-fitting covariance structure was selected based on information criteria and visualization of residual plots. Normality of residuals was assessed with the Shapiro–Wilk test. Non-normality was addressed with mathematical transformation. The models testing IC from the MTT and 30 min treadmill walk were repeated on treatment and included main effects for treatment and sequence, and included volunteer sex, BMI, and age as covariates. Covariates were removed by backward elimination if non-significant. The models built to test the glucose and insulin response curves were repeated on treatment and minute, and included main effects for treatment, minute, sequence, covariates, and a treatment*minute interaction term not subject to backward elimination. Similar models were built to test iAUC for both insulin and glucose; this is a single measure per 3 h period, and not repeated on time. Volunteers were included as a random-side effect term in all models. All models included Tukey post-hoc HSD correction when reporting group-wise differences. Data are presented as group means ± SEM, and statistical significance is considered when *p* < 0.05. The de-identified raw data is provided as Appendix A.

The microbiome data were visualized through non-metric multidimensional scaling (NMDS; Bray–Curtis similarity) and analyzed in Excel and statistical significance assessed through two-way analysis of similarities (ANOSIM), as well as Kruskal–Wallis and Mann and Whitney tests in the PAST software, v4.17 [25]. Stool samples were collected after subjects had already started to consume the test juices, gut microbiota dynamics were visualized and assessed both with the four sampling events as distinct groups, but also with comparison of EBJ and PL as pooled groups. In both cases, paired statistical tests were performed to ensure that statistical significance reflects subject-dependent trends.

## 3. Results

### 3.1. Study Volunteer Characteristics

The CONSORT diagram for this study is reported in Figure 1. Qualified participants were enrolled into the protocol from February of 2023 to April of 2024. Eighty-three prospective volunteers attended an information meeting, and 53 were lost to follow-up. Thirty prospective volunteers provided written informed consent. A total of six later declined to participate in the study, and 24 were screened for eligibility. Five volunteers were excluded for not meeting inclusion criteria, and 19 participants were randomized to the study protocol. One participant withdrew from the study for noncompliance, and 18 participants successfully completed the study protocol. Test beverage consumption compliance, assessed by the daily questionnaire and returned containers, was 100%. There were no written reports or verbal complaints of gastrointestinal distress or inability to consume the prescribed food and treatments, and therefore it is assumed both treatment beverages and background-controlled diets were well tolerated by participants. Baseline characteristics for the 18 included participants are reported in Table 3.

### 3.2. Indirect Calorimetry and Blood Glucose & Insulin Response

IC measurements of CHO, fat, EE, and RQ from the MTT and exercise challenges are reported in Table 4. One of the 18 participants was not successfully measured for IC outcomes, therefore measurements on 17 participants were available for statistical testing. There were significant treatment differences for average RQ and corresponding CHO and fat oxidation during the MTT. Average RQ was significantly lower with the EBJ treatment (0.86 vs. 0.89, EBJ vs. PL, respectively, *p* = 0.031). This corresponds to a significant decrease in CHO oxidation (25.6 vs. 30.0 g, EBJ vs. PL, respectively, *p* = 0.021) and a significant increase in fat oxidation (10.2 vs. 8.03 g, EBJ vs. PL, respectively, *p* = 0.038) with EBJ treatment compared to PL. There was no treatment effect in EE (202 vs. 200 kcal, EBJ vs. PL, respectively, *p* = 0.611). There was no significant effect of age, sex, or BMI.

IC measures were similar for the 30 min treadmill walk, but CHO and fat oxidation between treatments were only marginally statistically significant (*p* = 0.055, and 0.071, respectively). However, differences in average RQ between treatments maintained significance during the treadmill walk and in the same direction as the MTT comparison (0.89 vs. 0.91, EBJ vs. PL, *p* = 0.038). This indicates fat oxidation was increased during exercise with the EBJ treatment compared to PL. There was no effect of treatment sequence in any of the IC outcomes tested during the MTT or exercise challenges.

Serum glucose and insulin MTT-response curves and corresponding incremental area under the concentration curves are reported in Figure 2 and Table 5, respectively. There was not a significant treatment by time-point interaction for either glucose or insulin dose-response curves (*p* for interaction = 0.72, and 0.66, respectively). However, glucose iAUC was 24% lower following the EBJ meal challenge compared to PL (*p* = 0.041). Insulin iAUC was 9.9% lower following the EBJ meal challenge, but this effect was marginally significant (*p* = 0.062).

### 3.3. Fecal Microbiota

Using NMDS to visualize treatment effects in the gut microbiota profiles, no significant impact can be detected (ANOSIM, *p* > 0.5) for any group (Figure 3). Two individuals were consistently distinct (left side of the NMDS) from the other, a difference that was identified to be driven by higher abundance of *Prevotella*. NMDS without those individuals did not reveal any significant clustering.

Both the placebo and the EBJ induced comparable dynamics in the gut microbiota at the phylum level. The placebo treatment decreased the relative abundance of Bacteroidetes, and significantly increased the abundance of Firmicutes. The same dynamics, but less marked and not reaching statistical significance were observed with EBJ (Figure 4A). Both treatments also drove numerical increases in the average abundance of Actinobacteria, with large variance driven by a few individuals (Figure 4B). EBJ resulted in slightly higher Actinobacteria abundance (*p* = 0.16). Since volunteers had already been consuming the EB juice when baseline stool samples were obtained, we chose to pool study arms based on presence or absence of EBJ to better single out potential effects of anthocyanins on the gut microbiota [27]. We also removed the two outlier individuals from relative abundances analyses. This approach confirms that EBJ causes significant decrease of Bacteroidetes and significant increase of Firmicutes (Figure 4C), as well as numerical increase of Actinobacteria, whereas Proteobacteria and Verrumicrobia were not affected (Figure 4D).

Among the Firmicutes, the two most abundant taxonomic groups, *Faecalibacterium* and Ruminococcaceae (*Ruminococcuss* spp. and other related genera) were impacted by EBJ. *Faecalibacterium* relative abundance increased suggestively (*p* = 0.08), while the abundance of Ruminococcaceae genera was significantly (*p* = 0.02) increased (Figure 5A). On the other hand, low abundant lactic acid bacteria slightly decreased or remained stable (Figure 5B). Bacteroides, and to some extent Alistipes were the Bacteroidetes members negatively impacted by EBJ consumption (Figure 5C). Finally, many Actinobacteria genera and primarily Bifidobacterium’s relative abundances were increased by EBJ (Figure 5D).

## 4. Discussion

The objective of this study was to test the prospective effects of EBJ on glucose tolerance, energetics and substrate oxidation, and the gut microbiota. To ensure minimum variability between treatment groups, we employed a randomized, single-blind, placebo-controlled crossover design and successfully tested 18 study volunteers, which ensured statistical power to detect changes between groups. We also designed and employed a 2-day rotating menu that was employed across 4 days within each treatment period, which allowed for meaningful comparisons from the clinical testing, but more importantly, this allowed for comparison of the fecal microbiota profile between test beverages, as previous findings demonstrate measurable changes with controlled feeding periods as brief as 2 days [28]. A 3-week washout period was included in the study design to minimize the chance of carryover effects, which were not detected when comparing the two possible intervention sequences. The feeding experiment was informed by several earlier human feeding studies that demonstrate prospective benefits via the consumption of whole berries on clinical endpoints commonly associated with obesity and diabetes [12,13,29,30,31,32,33,34]. Here, we employed a similar testing regime with EBJ, as they are a more concentrated source of anthocyanins compared to other berries, and thus may achieve threshold doses with pragmatic serving sizes [35]. The dose was also informed by a pilot study that demonstrated consuming 177 g of EBJ per day for 1-week did not result in gastrointestinal complications [14], thus the present study provided 355 g of EBJ per day for the same duration, thereby providing over 720 mg of cyanidin-based anthocyanins per day [35]. As no ill effects were reported in the present study, this suggests 355 g daily EBJ consumption has an acceptable safety profile in adults, but this needs confirmation in controlled trials of longer duration.

Numerous previous rodent studies describe protective effects of anthocyanins, both from whole berry powders and isolated anthocyanins, against diet-induced obesity and associated metabolic aberrations [6,36,37,38,39,40,41,42]. Myriad mechanisms of action are measurably altered by berry anthocyanins, including themes related to digestion and absorption, the gut microbiome, and expression of regulatory proteins within metabolically active tissues which exert control over energy metabolism and storage [11]. It is difficult to suggest which mechanism deserves priority, but it has been proven that the improvements extend to EBJ, as our recent rodent study of diet-induced obesity recapitulates the findings noted with other berry sources of anthocyanins [15].

Most of the mechanisms of action highlighted above are difficult to study in small human feeding studies of short duration. However, prospective benefits can be investigated with the testing employed here: (1) meal tolerance testing, (2) indirect calorimetry, and (3) fecal microbiota profiling. Here, we describe for the first time changes in all three endpoints by EBJ. Improvement of glucose metabolism following a high carbohydrate meal has been demonstrated with other berries, including strawberries, blueberries, blackberries, and mixed-berry interventions [12,13,29,31]. The current findings suggest that this effect also extends to EBJ, as both a 24% reduction in the blood glucose response curve and 9% reduction in the insulin curve were observed. The same testing did not reveal any glucoregulatory relationship in our previous pilot study [14], which likely indicates the importance of controlling the background diet in the days leading to clinical testing, as was done here. The notable differences observed here require additional testing with better controls, which is described in the study limitations section below.

In-line with the significant changes in blood glucose following the MTT, the EBJ treatment significantly increased fat oxidation as evidenced by modulation of the RQ. We have previously noted this effect when healthy men with overweight or obesity were fed 600 g of blackberries daily for 1 week [12]. In both instances, these changes are independent of effects on energy expenditure. Like the observations with the blackberry study, the decreased RQ was not specific to the MTT, i.e., RQ remained significantly lower during 30 min of moderate physical activity. Similarly designed studies investigating IC and exercise outcomes with New Zealand blackcurrants also describe increases in fat oxidation [43,44,45,46,47,48,49]. Moreover, blueberry interventions incorporating physical activity demonstrate increased colonic absorption of gut-derived polyphenol metabolites, which likely possess greater bioactivity than parent compounds [50,51,52]. This suggests a synergistic relationship between physical activity and berry consumption. In summary, detection of these directional changes in RQ with acute feeding studies is suggestive of changing body composition in the long-term, which has been confirmed with longer intervention periods [53].

As is commonly the case [54], our short-term dietary intervention did not result in profound gut microbiota profile modulation. While classifying polyphenols and specifically anthocyanins as prebiotics is sometimes suggested [55,56,57], supporting evidence for such claims is partial at best. Subtler but sustained gut microbiota modulation should not be discounted in favor of obvious prebiotic properties, and here we discuss evidence for subtle gut microbiota modulation from EBJ’s anthocyanins. At the phylum level, EBJ induced slight remodeling of the gut microbiota, with increased Actinobacteria and Firmicutes relative to the Bacteroidetes. While the actual relevance of the Firmicutes to Bacteroidetes ratio to human health is hotly debated, it is also relatively clear that stimulating Firmicutes and Actinobacteria is generally more beneficial in the context of Western populations [58,59]. Interestingly, we observed slightly different trends at the phyla level in a rodent study using the same EBJ, with Actinobacteria markedly increased, and Bacteroidetes and Firmicutes virtually unchanged by EBJ [15].

The ideal balance of Bacteroidetes genera in the human gastrointestinal environment is debated, and likely dependent on habitual dietary habits and host background. Comparisons between human populations in distinct continents have consistently hinted at *Prevotella* as a beneficial genus, relative to *Bacteroides* as less desirable [60,61]. However, it is important to note that each of the three (*Prevotella*, *Bacteroides*, and *Alistipes*) have been associated with both positive and negative health properties in preclinical and human studies [62,63]. *Bacteroides* species have commonly been reported as important in the microbial degradation of anthocyanins, specifically the cleavage of sugar moieties [52]; so the decrease in relative abundance appears counterintuitive. Two hypotheses can explain this outcome: 1. a competitive inhibition of *Bacteroides* species lacking the ability to metabolize anthocyanins, or 2. a competition for anthocyanins from other gut microbiota members. *Alistipes* are not known for their ability to metabolize anthocyanins and followed the same trend, so the first hypothesis seems more likely.

In contrast with our rodent study [15], the relative abundance of *Akkermansia* was seemingly unaffected by EBJ consumption, and low compared to other human populations [64]. An expansion of *Akkermansia* was described as the main outcome in a human dietary intervention with an elderberry-based polyphenol extract [65]; however, this expansion is difficult to discern. *Akkermansia* expansion due to polyphenols and anthocyanins consumption has mostly been shown in rodent models [60,66,67]. Thus, this outcome may be specific to the animal gut environment, or present findings in human interventions are below a threshold dose or duration and therefore fail to elicit an effect.

In our rodent model study, we observed a decrease in lactic acid bacteria (LAB) with EBJ consumption [15], which is confirmed with the present human data. The only difference in both cases is EBJ intake, so this decrease can be directly attributed to anthocyanin consumption. Environmental LAB is known to degrade anthocyanins and other polyphenols [68], but such metabolic abilities are less evident for human gut-associated LAB members. In contrast, we report significant stimulation of the already abundant *Faecalibacterium* and Ruminococcaceae genera (including *Ruminococcus*). All these taxa are unequivocally considered as health beneficial [69] and consistently associated with complex polysaccharides, dietary fiber and resistant starch degradation [70,71,72], and the production of butyrate [73,74]. This is supported by a reported increase of butyrate producers in humans consuming wild blueberry powder [75]. In our rodent study, we observed a suggestive increase in the relative abundance of *Bifidobacterium* [15] mirroring what we observed here with human volunteers, though it should be noted that a subset of human volunteers were evidently responders as is commonly reported in other studies [76]. Notably, *Bifidobacterium dentium* has been shown to catabolize elderberry anthocyanins in vitro [77]. In addition, in the limited number of human dietary intervention with blueberries, *Bifidobacterium* increase was reported twice [78,79].

The present study has limitations. First, the study duration was limited to 1 week of treatment, and meaningful long-term effects of elderberry juice can only be confirmed in interventions spanning several months, which would also allow for inquiry into blood lipid homeostasis. Second, our study was not balanced for sex due to difficulties enrolling men in the Spokane, WA metropolitan area, therefore prospective benefits suggested by the current findings should be limited to overweight but otherwise healthy women. Third, while both challenge meals provided over 80 g of sugar, where the standardized clinical test for glucose tolerance provides 50–75 g sugar from a glucose solution, there was a 15 g difference in total sugar content between challenge meals. Therefore, our findings regarding any improvement in glucose tolerance and insulin sensitivity is suggestive but must be confirmed with additional testing where better control is placed on meal challenge design, including macronutrient and energy matching between meals. However, when research infrastructure is limited, the present study demonstrates an effective approach to controlled human feeding studies, nutritional science’s gold standard.

## 5. Conclusions

This is the first human clinical trial to demonstrate that daily consumption of EBJ for one week significantly increases gut microbial communities associated with health benefits for the host. Compared to placebo, EBJ significantly increased Firmicutes and Actinobacteria, and decreased Bacteroidetes phyla. At the genus level, EBJ increased *Faecalibacterium*, *Ruminococcaceae*, and *Bifidobacterium* and decreased *Bacteroides* and lactic acid-producing bacteria, indicating a positive response to EBJ. The findings also corroborate the positive effects of anthocyanin rich berry consumption on blood glucose homeostasis and fat oxidation, where a 24% reduction in the serum glucose area-under-the-curve, and 27% increase in fat oxidation were observed with the EBJ treatment. The findings suggest that anti-obesity effects of EBJ observed in translational research models do extend to humans. These observations need to be confirmed in longer duration trials that adequately investigate both sexes and broader age groups. Future studies should aim to investigate mechanisms of action using preclinical, clinical, and translational research models.

## Figures and Tables

**Figure 1 nutrients-16-03555-f001:**
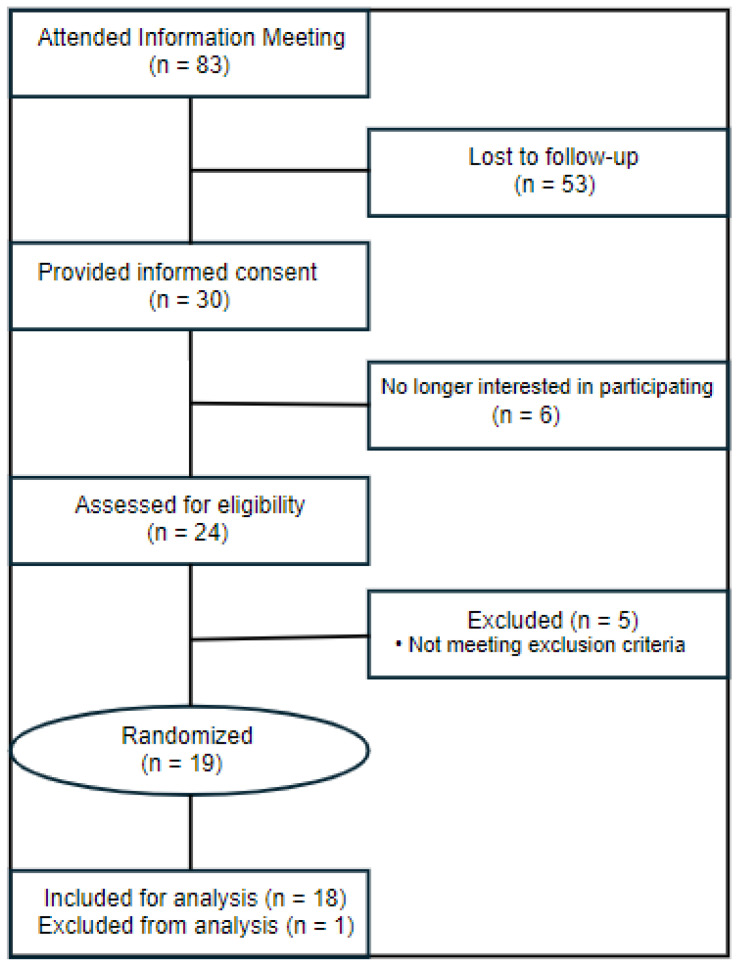
CONSORT (Consolidated Standards of Reporting Trials) diagram. One participant was excluded from analyses due to withdrawing from the study.

**Figure 2 nutrients-16-03555-f002:**
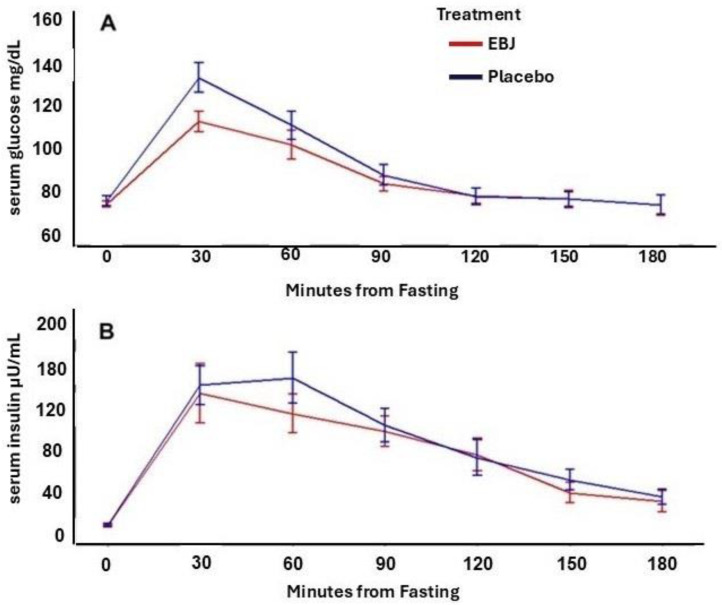
Blood glucose (**A**) and insulin (**B**) concentrations by time following a meal tolerance test including either elderberry juice or placebo treatments. EBJ, elderberry juice; PL, placebo. n = 18.

**Figure 3 nutrients-16-03555-f003:**
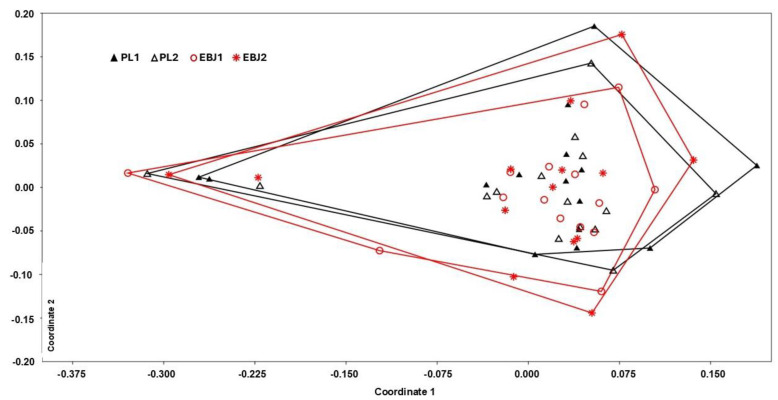
Non-metric multidimensional scaling (NMDS) plot comparing the four groups. EBJ1 or PL1, baseline fecal samples, EBJ2 or PL2, post-intervention fecal samples.

**Figure 4 nutrients-16-03555-f004:**
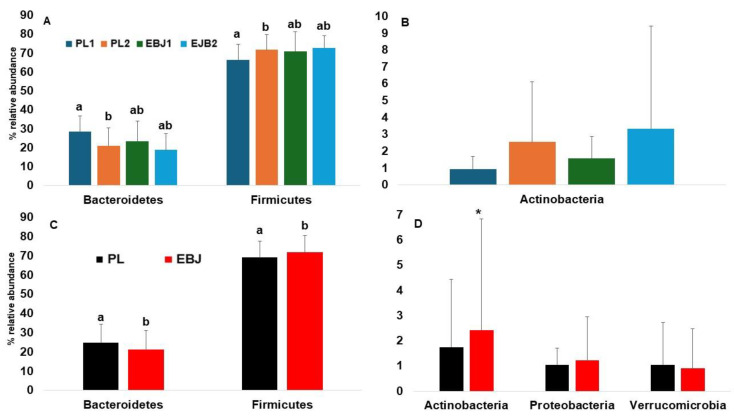
Changes in microbial population at phylum level; (**A**): changes in relative abundance of Bacteroidetes and Firmicutes comparing treatments EBJ1, EBJ2, PL1, and PL2, (**B**): changes in the relative abundance of Actinobacteria comparing treatments EBJ1, EBJ2, PL1, and PL2; (**C**): changes in relative abundance of Bacteroidetes and Firmicutes comparing combined EBJ and PL treatments; (**D**): changes in the relative abundance of Actinobacteria, Proteobacteria, and Verrucomicrobia comparing combined treatments EBJ and PL. EBJ1 or PL1, baseline fecal samples, EBJ2 or PL2, post-intervention fecal samples. Pair-wise comparisons without a common letter are significantly different, *p* < 0.05. Pair-wise comparisons with an asterisk are marginally different, *p* < 0.1.

**Figure 5 nutrients-16-03555-f005:**
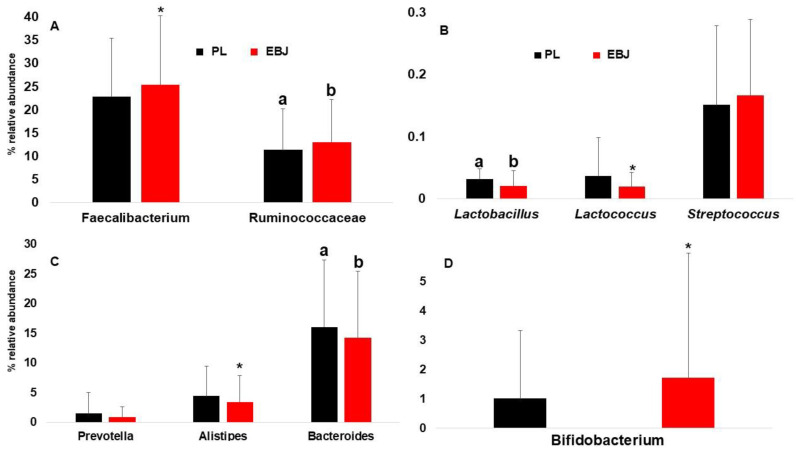
Changes in gut microbial population at genus level; (**A**): changes in relative abundance of *Faecalibacterium* and Ruminococcaceae comparing combined EBJ and PL; (**B**): changes in the relative abundance of *Lactobacillus*, *Lactococcus*, and *Streptococcus* comparing combined treatments EBJ and PL; (**C**): changes in relative abundance of *Prevotella*, *Alistipes*, and *Bacteroides* comparing combined EBJ and PL treatments; (**D**): changes in the relative abundance of *Bifidobacterium* comparing combined treatments EBJ and PL. EBJ, elderberry juice; PL, placebo. Pair-wise comparisons without a common letter are significantly different, *p* < 0.05. Pair-wise comparisons with an asterisk are marginally different, *p* < 0.1.

**Table 1 nutrients-16-03555-t001:** Anthocyanin content of elderberry juice.

Anthocyanin (mg)	Per 177.5 g Serving	Per 355 g Daily Dose
Cyanidin-3,5-diglucoside	41.4	82.8
Cyanidin-3-sambubioside-5-glucoside	56.3	112.6
Cyanidin-3-glucoside	0.1	0.1
Cyanidin-3-sambubioside	1.9	3.7
Cyanidin-3-rutinoside	0.4	0.8
Pelargonidin-3-glucoside	0.94	1.9
Cyanidin-based anthocyanin	10.3	20.6
Delphinidin-3-rutinoside	0.4	0.9
Cyanidin-3-(Z)-p-coumaroyl-sambubioside-5-glucoside	1.2	2.3
Cyanidin-3-p-coumaroyl-glucoside	1.8	3.5
Petunidin-3-rutinoside	2.6	5.2
Cyanidin3-(E)-p-coumaroyl-sambubioside-5-glucoside	258.8	517.6
Cyanidin-3-p-coumaroyl-sambubioside	1.6	3.1
Total measured anthocyanins	377.5	755

**Table 2 nutrients-16-03555-t002:** Macronutrient and energy content of the meal-based tolerance test foods.

	Quantity (g)	Protein (g)	CHO ^1^ (g)	Sugar (g)	Fat (g)	Energy (kcals)
Elderberry Treatment						
Waffle	80	4.6	34.3	4.6	5.7	205.7
Syrup	80	-	72	58.7	-	293.3
Elderberry Juice	355	-	23.7	23.7	-	95
Total	515	4.6	130	87	5.7	594
Placebo Treatment						
Waffle	80	4.6	34.3	4.6	5.7	205.7
Syrup	80	-	72	58.7	-	293.3
Placebo	355	-	39.2	39.2	-	156.8
Total	515	4.6	145.5	102.5	5.7	655.8

^1^ CHO, carbohydrate.

**Table 3 nutrients-16-03555-t003:** Baseline participant characteristics ^1^.

Characteristic	Mean ± SEM
N (sex)	15 (f) 3 (m)
Age (years)	40.6 ± 3.7
Weight (kg)	81.9 ± 3.7
BMI (kg/m^2^)	29.12 ± 0.7
Waist Circumference (cm)	96.9 ± 2.3
Total Cholesterol (mg/dL)	178.9 ± 7.3
LDL (mg/dL)	100.3 ± 5.9
HDL (mg/dL)	61.8 ± 3.7
TG (mg/dL)	92.6 ± 8.4
Systolic Blood Pressure (mm Hg)	121.6 ± 3.0
Diastolic Blood Pressure (mm Hg)	81.7 ± 2.4
Glucose (mg/dL)	90.9 ± 1.6

^1^ Total cholesterol, LDL, HDL, TG, and glucose tests are measured in 12 h fasted serum samples.

**Table 4 nutrients-16-03555-t004:** Postprandial and exercise substrate oxidation, EE, and mean RQ in adults following a meal tolerance test including either elderberry juice or placebo treatments.

	EBJ	PL	
	Mean ± SEM	Mean ± SEM	*p*
3-h meal challenge			
CHO (g)	25.6 ± 1.76	30.0 ± 1.24	0.021
Fat (g)	10.2 ± 1.00	8.03 ± 0.73	0.038
EE (Kcal)	202 ± 5.16	200 ± 5.13	0.611
Mean RQ	0.858 ± 0.012	0.886 ± 0.009	0.031
30-min treadmill walk			
CHO (g)	20.6 ± 0.879	22.0 ± 1.17	0.055
Fat (g)	4.99 ± 0.333	4.32 ± 0.346	0.071
EE (Kcal)	133 ± 3.77	132 ± 4.91	0.815
Mean RQ	0.891 ± 0.006	0.905 ± 0.007	0.038

EBJ, elderberry juice; PL, placebo; CHO, carbohydrate oxidation; Fat, fat oxidation; EE, energy expenditure; RQ, respiratory quotient. *n* = 17.

**Table 5 nutrients-16-03555-t005:** The 180 min serum glucose and insulin iAUC levels in adults following a meal tolerance test including either elderberry juice or placebo treatments.

	EBJ	PL	*p*
Glucose iAUC (mg·minute per dL)	2929 ± 479	3746 ± 397	0.041
Insulin iAUC (µU·minute per mL)	12,849 ± 2009	14,258 ± 1995	0.062

iAUC, incremental area under the curve; EBJ, elderberry juice; PL, placebo. n = 18.

## Data Availability

De-identified research data from measures of indirect calorimetry, blood glucose, and blood insulin are available in Appendix A described above. The research data from the 16S_rRNA fecal microbiota sequencing are readily accessible at https://www.ncbi.nlm.nih.gov/bioproject/1163410.

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
