# Peer review of "A One-Week Elderberry Juice Intervention Augments the Fecal Microbiota and Suggests Improvement in Glucose Tolerance and Fat Oxidation in a Randomized Controlled Trial"

_nutrients, 2024, doi:10.3390/nu16203555_

Round 1
Reviewer 1 Report
Comments and Suggestions for Authors
The authors describe a randomised, placebo-controlled study of the effects on glucose regulation, substrate oxidation and faecal microbiota, of a one-week elderberry juice intervention in a cohort of adults without chronic illness but with a BMI > 25 kg/m2. The study is well designed, although a larger sample, studied for longer, with a better sex ratio, would be important in any follow-up study. The statistical techniques are appropriate and the results are well illustrated.
There are no major amendments required. I should like to suggest the two following minor amendments to what is otherwise an excellent submission.
1. The title describes the cohort as being healthy. I wonder if mention should be made here of their relatively high body mass (or weight)?
2. In the abstract, I suggest adding the units for the BMI (immediately after "25").
Author Response
We are grateful to the reviewer for their helpful feedback. We agree with the points they raise regarding follow-up studies, especially longer durations and better sex ratios. Our edits made in response to their requested amendments are as follows:
- We agree the title is problematic in this regard. We would prefer to keep specific details of study volunteer weight category constrained to the abstract and paper and therefore are removing any mention of “healthy” from the title as well.
- In the abstract, on line 26, we have now included “Overweight or obese adults, (BMI > 25 kg/m2)” so interested readers are immediately informed of study volunteer characteristics starting in the abstract and then in full detail in methods section 2.1.
Thank you for improving our report; we hope the reviewer finds these edits acceptable.
Reviewer 2 Report
Comments and Suggestions for Authors
The paper is very interesting. Authors focused on significant problem of obesity which can be partly solved by using an appropriate microflora or supporting the development of an existing one with a suitable diet enriched with natural active compounds. On the whole, paper is well prepared and aim of presented studies is suitable this Journal. Please see my small suggestions:
Abstract: informative, include most important study results
Introduction: based on up-to-date references, informative and readable
Methods: all methods are adequate to the aim of the studies. Authors decided to perform MTT, blood glucose and plasma insulin analysis as well as determine the effect of EBJ on the gut microbiome composition. Studies were well planned. Human partcipants were well recruited and appropriate criteria for the selection of study groups were applied. Statistical analyisis is provided.
Results: All important information are presented explicitly and readable. Please improve quality of Figures 2,3 and 4 (axes are not very readable).
Discusion: well prepared. The authors have discussed the results of the study very thoroughly based on the available literature.
Conclusions: can be improved. Please provide more detail information (quantitative reference to the results obtained)
Author Response
Author response:
We thank the reviewer for the kind feedback and helpful suggestions. Feeding studies are very difficult to conduct and we appreciate that the reviewer acknowledges our careful planning and sees the value in our findings. Our edits made in response to their guidance are as follows:
- We agree that the readability of figures 2-4 are problematic for prospective readers. We have edited all three figures in an effort to improve readability.
- We agree that our conclusions can be improved. We have now included quantitative references to our results. In this section, we have added “Compared to placebo, EBJ significantly increased Firmicutes and Actinobacteria, and de-creased Bacteroidetes phyla. At the genus level, EBJ increased Faecalibacterium, Ruminococ-caceae, and Bifidobacterium and decreased Bacteroides and lactic acid-producing bacteria, indicating a positive response to EBJ” on lines 507-510. We have also included “The findings also corroborate the positive effects of anthocyanin rich berry consumption on blood glucose homeostasis and fat oxidation, where a 24% reduction in the serum glucose area-under-the-curve, and 27% increase in fat oxidation were observed with the EBJ treatment” on lines 510-513.
Thank you for your time and for improving our report; we hope the reviewer finds these edits acceptable.